# The Effects of a Trace Amount of Manganese and the Homogenization on the Recrystallization of Al–7Mg–0.15Ti Alloys

**DOI:** 10.3390/molecules26010168

**Published:** 2020-12-31

**Authors:** Yang-Chun Chiu, Tse-An Pan, Guan-Ming Chen, Xin-Cheng Jiang, Hui-Yun Bor, Yu-Chih Tzeng, Sheng-Long Lee

**Affiliations:** 1Institute of Materials Science and Engineering, National Central University, Jhongli 32001, Taiwan; albert77918@gmail.com (Y.-C.C.); peterpan.ck@gmail.com (T.-A.P.); ming821224@gmail.com (G.-M.C.); s0021232@hotmail.com (X.-C.J.); hohh@csnet.gov.tw (H.-Y.B.); 2Department of Mechanical Engineering, Minghsin University of Science and Technology, Hsinchu 30401, Taiwan; 3Department of Power Vehicle and Systems Engineering, Chung-Cheng Institute of Technology, National Defense University, Taoyuan 334, Taiwan; a0932467761@gmail.com

**Keywords:** Al–Mg alloy, manganese, two-stage homogenization, Al_4_Mn dispersoids, recrystallized grains, mechanical properties

## Abstract

The aim of this study is to explore the effects of Manganese addition and homogenization treatment on the microstructures and mechanical properties of the Al–7Mg–0.15Ti (B535.0) alloy. The optical microscopy, electrical conductivity measurements, transmission electron microscopy, scanning electron microscopy (SEM + EBSD), as well as Rockwell hardness and tensile tests, were exploited for this purpose. The main objectives are to refine the grain size, inhibit grain growth in the annealed state, and enhance the mechanical strength of the alloy. The results show that the addition of manganese to the Al–7Mg–0.15Ti alloys refined the as-cast and recrystallized grains of the alloys. During the homogenization process, Al_4_Mn high-temperature stable dispersoids were precipitated in the aluminum matrix. After annealing, the Al_4_Mn particles blocked the movement of grain boundaries during the growth of the recrystallized grains and inhibited grain growth. Consequently, the annealed alloys showed grain refinement and dispersion strengthening. The Al_4_Mn dispersoids of the alloys with manganese added were smaller and denser after a two-stage homogenization process compared to those that underwent a one-stage homogenization process. By contrast, for the alloys without the addition of manganese, the recrystallized grains showed normal growth after annealing, and different homogenization processes had no significantly different effects.

## 1. Introduction

The 500 series aluminum–magnesium alloy with magnesium atoms as the main alloying element is a cast, non-heat-treated aluminum alloy with good solid solution-strengthening characteristics, and its strength increases with increasing magnesium content [1]. Other trace elements such as manganese, titanium, and zirconium can be added to the alloy to obtain Al_4_Mn, Al_3_Ti, and Al_3_Zr second-phase precipitates in the aluminum–magnesium matrix. These grains have high-temperature thermal stability [1], which can inhibit the growth of alloy grains. The precipitates spread throughout the aluminum matrix to enhance the dispersion strengthening of the alloy [1]. With processing, it can be made into a forged alloy. This forged aluminum–magnesium alloy is significantly strengthened as a result of processing, which further improves the strength of the alloy. Due to its good corrosion resistance and workability [1], it has been widely used in ship structures, car body structures, electronic components, and other materials [1].

Adding manganese to the aluminum–magnesium alloy can generate Al_6_Mn at the grain boundaries during solidification to refine the as-cast grains of the alloy and achieve fine-grain strengthening [2,3]. Al_6_Mn can also form an Al_6_(Mn,Fe) phase to reduce the adverse effect of iron on the ductility of the aluminum alloy. At the same time, during the homogenization, the dispersed Al_4_Mn phase [4] is precipitated to achieve dispersion strengthening. Al_4_Mn has been shown to coarsen with the increase in homogenization time and temperature, and thereby promote the hot workability [2]. The precipitates of manganese can hinder the movement of dislocations and the grain boundary, thus increasing the strength of the alloy. In addition, the elongation will not be greatly reduced [5]. When the alloy is subjected to cold working, a large number of dislocations accumulate around the precipitated phase, making its stored energy higher than that in the matrix. Therefore, in the annealing process, Al_4_Mn can promote recrystallization and nucleation, and limit grain growth [6,7]. Mn-containing forged aluminum–magnesium alloys can increase the alloy recrystallization temperature. If the annealing temperature is sufficiently low, the alloy can still retain the fibrous structure after the annealing process [1]. On the other hand, if the manganese-containing aluminum–magnesium alloy is processed at a very high temperature, and with longtime annealing, secondary recrystallization occurs [7].

The homogenization treatment is mainly used to eliminate micro-segregation of the as-cast alloy. Due to the fast cooling rate during casting, transition metals such as Fe, Mn, and Zr are dissolved in the aluminum alloy to form the supersaturated solid solution. The homogenization treatment can result in the transition metals and the aluminum precipitating as intermetallic compounds. The precipitation phase of the trace elements varies depending on the temperature and time of the homogenization treatment [8,9]. According to the Zener Pinning theory, the finer or denser the grains precipitated in the dispersed phase, the more that recrystallization and grain growth are suppressed during annealing [10,11]. Therefore, when the Al–Mg–Mn alloy changes the precipitation morphology of the dispersed phase after the two-stage homogenization, there is greater inhibition of recrystallization during annealing and the grains are smaller than those obtained after the one-stage homogenization [12]. In the two-stage homogenization, the temperature of the first stage of homogenization affects the mechanical properties of the alloy. In the first stage of the homogenization, using a high temperature (such as 400 °C) can result in better mechanical properties than using low temperature (such as 200 °C) [13].

According to Considère’s criterion and Hart’s criterion [14], when the alloy deforms, if the hardening rate is not less than the true stress, the alloy will deform uniformly. That is, the high work hardening rate precludes the occurrence of necking. On the contrary, if the hardening rate is less than the true stress, necking will occur, and the alloy will fracture. Therefore, the strength and ductility of the alloy, with a high work hardening rate, may be increased with the increase in the solute atomic mass.

In early research work [4,6,15], it was found that changing the homogenization heat treatment conditions of the Al–4.5Mg–Mn alloy significantly affected the morphology of Al_4_Mn precipitates, the subsequent hot workability, and annealing grain size. In this study, B535.0 (Al–7Mg–0.15Ti) alloys were used and a trace amount of manganese (0.8 wt%) was added to the alloys. Through plastic working, the original casting alloy was made into a forged alloy. Two homogenization processes were conducted for the heat treatment to evaluate the effects of the annealing time on the recrystallization of aluminum–magnesium alloys through observation and analysis of the microstructure, and mechanical property testing [16,17]. As far as the collected documents are concerned [18,19,20,21], there is no similar research report.

## 2. Results and Discussion

The purpose of this study was to inhibit the recrystallization of the commercial B535.0 (Al–7Mg–0.15Ti) aluminum alloy and enhance its mechanical properties. Certainly, the above designed and experimental alloys could not fully or satisfactorily replace the functions of the B535.0 alloy. For one thing, the tensile test could not elucidate all the mechanical properties. For another, the chemical and physical properties of the alloys require further investigation. However, this study might offer a valuable reference for future researchers.

### 2.1. Microstructure Analysis

Figure 1a shows the as-cast optical microstructure of Alloy A (0Mn), and the crystal grains appeared to be equiaxed. The diameter of the crystal grains measured by the intercept method was about 180 μm, and it was found that there were many dendrites produced by non-equilibrium cooling in the crystal grains. The microstructures of Alloy A after the one-stage homogenization and the two-stage homogenization processes, respectively, are shown in Figure 1b,c. The dendrites in the as-cast state were clearly observed. Whether the alloy was subjected to the one-stage homogenization or the two-stage homogenization process, the dendrites were eliminated. The grain size after the homogenization did not differ greatly from that of the as-cast grain. Figure 1d shows the as-cast optical microstructure of the Mn-containing Alloy B (0.8Mn). The crystal grains were equiaxed dendritic grains with a diameter of about 98 µm. The microstructures after the homogenization are shown in Figure 1e,f. The dendrites in the as-cast state were fully eliminated after both homogenization processes. The grain size after the homogenizations was also similar to the size of the as-cast grain.

It is evident that the grains of Alloy B (0.8Mn) were smaller than those of Alloy A (0Mn), whether in the as-cast state or after the homogenization. Therefore, the addition of Mn resulted in grain refinement of the casting grains in the aluminum–magnesium alloys [3]. In addition, the Al_4_Mn dispersed phase that precipitated during the homogenization process suppressed the growth of crystal grains. Similarly, the two kinds of homogenization heat treatments effectively eliminated the micro-segregation present in the as-cast alloy and achieved homogenization.

The Mn-containing as-cast Alloy B (0.8Mn) was observed with SEM-BEI (Backscattering Electron Image of Scanning Electron Microscope). As shown in Figure 2a, the Fe-containing Al_6_Mn crystallized phase existed at the grain boundary [3]. However, no other crystalline phases were observed in the aluminum matrix. After Alloy B (0.8Mn) was subjected to the one-stage homogenization heat treatment (530 °C for 12 h) and the two-stage homogenization heat treatment (430 °C for 8 h + 530 °C for 10 h), respectively, many dispersoids were precipitated in the grains, as shown in Figure 2b,c. By contrast, Figure 2d shows that after the two-stage homogenization heat treatment of Alloy A (0Mn), no dispersoids were observed in the grains.

These precipitates in the grains of Alloy B (0.8Mn) after the homogenization heat treatment were analyzed by TEM diffraction analysis. They were hexagonal close-packed Al_4_Mn particles of the high-temperature stable dispersoids. As shown in Figure 3a,b, their sizes were approximately 400 and 250 nm, respectively. The high-temperature stable precipitated dispersoids of Alloy B (0.8Mn) after the two-stage homogenization heat treatment were smaller and denser than those after the one-stage homogenization heat treatment. That is, in the one-stage homogenization heat treatment (530 °C for 12 h), the resulting nucleated Al_4_Mn dispersoids were coarser due to the higher temperature. Comparatively, in the first stage (the low-temperature homogenization heat treatment process (430 °C for 8 h)) of the two-stage homogenization (430 °C for 8 h + 530 °C for 10 h), a large amount of smaller Al_4_Mn particles nucleated and precipitated. Despite the high temperature (530 °C) used in the second stage of the homogenization heat treatment process, it did not cause obvious coarsening of the Al_4_Mn particles [11].

Figure 3c shows the as-cast Alloy B (0.8Mn) without any homogenization treatment. The phase containing Al_4_Mn particles was present in small amounts in the as-cast aluminum matrix. The coarse crystallized Al_6_Mn particles during casting were observed only at the grain boundaries. As for Alloy A (0Mn), after the homogenization heat treatment, no precipitated phases were observed in the aluminum matrix, as shown in Figure 3d.

As shown in Figure 4a, the microstructure of Alloy A (0Mn), after the one-stage homogenization heat treatment, 420 °C hot rolling (80%), and 400 °C full annealing for 2.5 h, consisted of fully recrystallized equiaxed grains. The size of the grain, as calculated by the intercept method, was approximately 60 µm. The microstructure of Alloy A (0Mn) after the two-stage homogenization heat treatment and full annealing showed no difference from that in Figure 4b. In Figure 4b, the residual direction of hot rolling could also be observed roughly, mainly because the amount of hot working per pass was less than 10%. Before each pass of the hot rolling process, the alloy was preheated at 420 °C for 5 min, so the accumulated stored energy in the alloy was very low during the process. Throughout the annealing, the orientation of the crystals after hot rolling was retained.

Figure 4b,c show the microstructures of Alloy B (0.8Mn) after the one-stage and the two-stage homogenization heat treatments and full annealing, respectively. Similarly, both had a completely recrystallized equiaxed grain morphology. However, the crystal grains of Alloy B (0.8Mn) were evidently finer than those of Alloy A (0Mn). Their grain sizes were approximately 38 and 29 µm, respectively. This was due to the Al_4_Mn dispersoids in Alloy B. The findings of previous studies [5,9] confirmed that Al_4_Mn dispersoids became nucleation points for annealing recrystallization. That is, Al_4_Mn dispersoids promoted recrystallization. As shown in Figure 3a,b, for Alloy B (0.8Mn) after the two-stage homogenization heat treatment, the high-temperature stable precipitated Al_4_Mn dispersoids were denser than those after the one-stage homogenization heat treatment and became the second crystal nucleation points. Moreover, the finer and denser Al_4_Mn dispersoids more effectively limited the movement of the grain boundaries. As a result, the grains of Alloy B (0.8Mn) after the two-stage homogenization heat treatment were the finest, while those of Alloy A (0Mn) were the coarsest. Figure 3c shows the grains of Alloy B (0.8Mn) after the two-stage homogenization heat treatment. The Al_4_Mn dispersoids were shown-using TEM to exist at the annealed grain boundaries, indicating their inhibition of grain boundary movement.

After the two-stage homogenization heat treatment, the Al–7Mg–0.15Ti alloys were hot rolled, fully annealed, and then subjected to 60% room-temperature cold rolling. The microstructures of Alloy A (0Mn) and Alloy B (0.8Mn) are shown in Figure 5a and Figure 5b, respectively. The alloys had plastic deformation after cold working at room temperature, so the crystal grains all had a slender, plastically deformed structure. After careful observation, it was found that Alloy B (0.8Mn) was denser than Alloy A (0Mn). It showed that the presence of Al_4_Mn dispersoids suppressed dislocations and increased the grain strengthening in Alloy B (0.8Mn). When the alloy was subjected to the one-stage homogenization heat treatment, the 60% cold-rolled microstructures of Alloy A (0Mn) and Alloy B (0.8Mn) were very similar, as shown in Figure 5a and Figure 5b, respectively. It was reasonable to speculate that after the one-stage homogenization, the density of the 60% cold-rolled Alloy B (0.8Mn) structure should be slightly lower than that after the two-stage homogenization; however, the difference could not be observed under the optical microscope (OM, Olympus BX60M, Shinjuku, Tokyo, Japan).

Figure 6 shows the microstructures of the Al–7Mg–0.15Ti alloy after cold working and annealing at 400 °C by means of EBSD analysis. Alloy A (0Mn), after the one-stage homogenization heat treatment, was completely recrystallized in only 0.5 h. As shown in Figure 6a, the crystal grains had an equiaxed spherical structure. By means of EBSD software analysis, the grain size was approximately 45 µm. As the annealing time increased to one hour, the crystal grains grew significantly. Their size was approximately 53 µm, as shown in Figure 6b. In addition, when Alloy A (0Mn) underwent the two-stage homogenization heat treatment, its degree of recrystallization was the same as that obtained after the one-stage homogenization heat treatment. The alloy grains also grew significantly as the annealing time increased. After the homogenization heat treatment of Alloy A (0Mn), because there were no Al_4_Mn dispersoids precipitated in the aluminum matrix, annealing at 400 °C neither promoted recrystallization nucleation nor inhibited grain growth. Therefore, when the annealing times were the same, there was no difference in the sizes of the crystal grains whether the alloys underwent the one-stage or two-stage homogenization heat treatment. As shown in Figure 6b,d, the crystal grains had normal grain growth with the increase in the recrystallization time, and were equiaxed, as shown in Figure 6a,b.

Figure 7 shows the microstructures of Alloy B (0.8Mn) after cold working and annealing at 400 °C by means of EBSD analysis. Similarly, Alloy B (0.8Mn) that underwent the one-stage homogenization heat treatment was completely recrystallized in 0.5 h. As shown in Figure 7a, average grain size was only approximately 30 µm, as determined by EBSD software analysis. As the annealing time increased to 1 h, the recrystallized grains had no obvious growth, as shown in Figure 7b. In addition, after the two-stage homogenization heat treatment, the degree of recrystallization was the same as that after the one-stage homogenization heat treatment; however, the average grain size was smaller (only approximately 26 μm) as shown in Figure 7c,d, and the crystal grains did not grow with the increase in the annealing time.

By measuring the change in conductivity (%IACS), the microstructures and precipitation states of the aluminum alloys were determined. The concentration of point defects had the greatest influence on the conductivity of the alloys. The way in which point defects occurred in the aluminum–magnesium alloys was mainly because of crystal lattice distortion caused by the solid dissolution of Mg and Mn atoms in the aluminum matrix and the dislocation pile-up caused during the processing. Table 1 shows the electrical conductivities of the alloys after various processing conditions.

After 60% cold rolling of Alloy A (0Mn), the crystal grains became the fibrous structure due to plastic deformation, as shown in Figure 6a. Large numbers of point defects and dislocations accumulated inside the crystal grains, resulting in a lower conductivity of the alloy in the cold-rolled state than in the recrystallized state. Even after different homogenization heat treatments, the total amount of solid solution elements were similar, so the electrical conductivity did not significantly differ. It can be seen from Table 1 that the conductivity (%IACS) of the cold-worked Alloy A (0Mn) was approximately 26.4%. By contrast, the conductivity of manganese (6.9 × 105 Ω^−1^ m^−1^) was not as good as that of aluminum (3.8 × 107 Ω^−1^ m^−1^), so the conductivity of Alloy B (0.8Mn) was lower than that of Alloy A (0Mn). The conductivity of Alloy B (0.8Mn) was approximately 23.3. There was a difference of approximately 12% between the two alloys.

After the cold-rolled alloys were recrystallized and annealed, large numbers of dislocations and point defects were eliminated, resulting in an increase in the conductivity of the alloys. The electrical conductivity (%IACS) of Alloy A (0Mn) and Alloy B (0.8Mn) were approximately 27.4 and 24, respectively. It can also be seen from Table 1 that when the annealing time was increased to 1 h, the conductivity of the alloys did not change significantly. This means that after annealing for 0.5 h, all the point defects and dislocations in the alloys were almost in equilibrium. The factors affecting the conductivity of the alloys were fully eliminated, so even if the alloys were annealed for a longer time, the conductivity of the alloy could not be improved further. This result was consistent with the above-mentioned micro-structural observation in Figure 6 and Figure 7.

Table 1 also shows that although the homogenization had little effect on the conductivity of Alloy B (0.8Mn) in the recrystallized state, with a change of approximately 1%, it was still higher than that of Alloy A (0Mn). This might be because the two-stage homogenization heat treatment precipitated denser Al_4_Mn high-temperature stable dispersoids than those of the one-stage homogenization heat treatment. As a result, the alloys that underwent the two-stage homogenization heat treatment had lower conductivities.

### 2.2. Mechanical Properties Tests

The results of the mechanical properties tests of Al–7Mg–0.15Ti alloys in different states are summarized in Table 2 and Figure 8. It can be seen from Table 2 and Figure 8a that whether Alloy B (0.8Mn) was subjected to the one-stage or two-stage homogenization heat treatment, the hardness, strength, and ductility were significantly improved to approximately 5.9% to 30% higher, respectively, than those of the Alloy A (0Mn). It is worth noting that after the two-stage homogenization, the values of the hardness, strength, and ductility of Alloy B (0.8Mn) were approximately 2.1–6.4% higher than the corresponding values after the one-stage homogenization. However, Alloy A (0Mn) had no such differences. This result can be explained by the changes in the microstructures discussed in the previous section. That is, Alloy B (0.8Mn) had the finest and densest Al_4_Mn thermally stable phase dispersoids after the two-stage homogenization. As a result, it had the best grain dispersion strengthening and the highest processing strengthening, as shown in Figure 5b. Furthermore, it had the highest mechanical strength and hardness in the 60% cold-working condition.

Despite the homogenization heat treatment that the recrystallized Alloy A (0Mn) was subjected to, if the annealing time was the same, the mechanical properties were similar. After annealing at 400 °C for 0.5 h as shown in Figure 8b, work hardening was fully eliminated. As a result, the hardness and strength of the alloy were significantly reduced for the 20% cold-rolled state compared to those of the 60% cold-rolled state, but the ductility greatly increased from approximately 7% to 20%. Similarly, as the annealing time increased to 1 h, it can be seen from the microstructures in Figure 6 that the alloy grains grew and became coarser. Consequently, the grain strengthening effect was reduced, and the mechanical properties of the alloy also changed slightly. The strength of the alloy decreased by approximately 1.7%, but its ductility increased from approximately 20.5% to 22.5%.

After the different homogenizations and annealing at 400 °C for 0.5 h, the processing strengthening of the recrystallized Alloy B (0.8Mn) was also fully eliminated as with that of Alloy A (0Mn). Its strength was greatly reduced by approximately 20%, but its ductility was greatly increased from approximately 8% to about 18%, compared to the strength and ductility in the 60% cold-rolled state. However, unlike Alloy A (0Mn), when the annealing time was increased to 1 h as shown Figure 8c, the mechanical properties of the alloy differed little from those when the annealing time was 0.5 h. These phenomena might be due to the fact that the dispersion strengthening of Al_4_Mn’s high-temperature stable dispersoids and the fine grain strengthening of the crystal grains did not change significantly despite the increased annealing time.

In addition, it can be seen in Table 2 and Figure 8d that the hardness and strength of the recrystallized Alloy B (0.8Mn) after the two-stage homogenization were approximately 2.2% and 6.8% higher than the corresponding values after the one-stage homogenization. This was also due to the dispersion strengthening of the fine and dense Al_4_Mn dispersoids of Alloy B (0.8Mn) through the two-stage homogenization, as shown in Figure 5b.

Table 2 shows the comparison of the changes in the ductility (EL%) of Alloy A (0Mn) and Alloy B (0.8Mn) in each state. It can be found that in the recrystallized state at 400 °C, the ductility of the alloy decreased, while its strength increased. That was a common characteristic of the material. That is, in comparison with Alloy A (0Mn), the ductility of Alloy B (0.8Mn) decreased when its strength increased. However, the ductility of the 60% cold-rolled alloy was different. Its ductility increased with the increase in its strength, which was different from the characteristics of the general materials. This result might be related to the strain hardening rate of the alloy during the tensile test. According to Considère’s criterion and Hart’s criterion, if the alloy has a high strain hardening rate, the ductility of the alloy will increase as its strength increases. With the above description of the strengthening mechanism of the Al–7Mg–0.15Ti alloy, it was shown that the 60% cold-rolled Alloy B (0.8Mn) had more strengthening mechanisms than those in the recrystallized state. The Al–7Mg–0.15Ti alloy also benefited from solid solution, dispersion, and processing strengthening, resulting in a higher work hardening rate during the tensile test.

## 3. Materials and Methods

After melting 99% pure aluminum ingots in a resistance crucible furnace at 75 °C, the selected amounts of 99.9% pure magnesium, the Al–75Ti master alloy, and the Al–75Mn master alloy were added. When all the reagents had melted, they were fully homogenized and degassed with pure argon gas for 30 min. After standing for 5 min, the melted solution was cast in a preheated (300 °C) metal mold with a size of 125 mm × 100 mm × 25 mm. According to the different manganese contents, Alloy A (0Mn): (Al–7Mg–0.15Ti) and Alloy B (0.8Mn): (Al–7Mg–0.8Mn–0.15Ti) were produced. The compositions of the experimental alloy samples were analyzed using an optical emission spectrometer (OES, Agilent 725). The analysis results are shown in Table 3.

Alloy A and Alloy B were placed in an air furnace and underwent the one-stage homogenization heat treatment (530 °C for 12 h) and the two-stage homogenization heat treatment (430 °C for 8 h + 530 °C for 10 h), respectively. After the homogenizing heat treatment, the alloys were quenched to room temperature, and hot rolled at 420 °C. Before each pass of hot rolling, preheating (420 °C for 5 min) was carried out, and the rolling amount of each pass was 0.25 mm. The alloys were rolled from 25 mm to 5.2 mm (80%). After thermal processing, complete annealing was performed at 400 °C for 2.5 h. Subsequently, the alloy pieces were cold-rolled from 5.2 mm to 2.1 mm (60%) at room temperature. After cold rolling, the test pieces were placed in an air furnace at 400 °C for recrystallization annealing heat treatment for 0.5 h and 1 h.

Under constant voltage (20 V), Barker’s reagent (5 mL HBF_4_ + 200 mL H_2_O) was used for anodizing to make optical micrographs. According to ASTM (American Society for Testing and Materials) specifications, Electrolyte I-1 (80 mL ethanol + 14 mL H_2_O + 6 mL HClO_4_) was applied at 20 voltages and the electrolytic polishing was performed at room temperature for 5 s to produce EBSD (Electron Backscatter Diffraction) test pieces. The Transmission Electron Microscope (TEM) test pieces were made by twin-jet electro polishing under constant voltage (33 mL HNO_3_ + 67 mL methanol) at −20 °C. The alloy microstructures were observed with an optical microscope (OM, Olympus BX60M, Shinjuku, Tokyo, Japan). A field emission gun transmission electron microscope (TEM, FEI Tecnai G2 F20, Hillsboro, OR, USA) was used to observe the morphology of the precipitated phase. A field emission scanning electron microscope (JEOL, JSM-7800F Prime) installed with EBSD was used to analyze the alloy grain boundaries and angles. Tango auxiliary analysis software was used to calculate the alloy grain size and the degree of recrystallization. The alloy conductivity (IACS%, International Annealing Copper Standard) was measured using a Sigmascope SMP10 Conductivity Meter. The alloy hardness was measured using the Rockwell Hardness Scale F (HRF). A 10-ton MTS closed-loop and hydraulic servo-controlled universal testing machine was used to perform room temperature tensile tests on cold-rolled and annealed alloys. The initial tensile rate was 0.2 mm/min. When the strain reached 1%, the tensile rate was increased to 2 mm/min. The test piece was in accordance with the ASTM tensile test bar specification, as shown in Figure 9 [22]. For the test of hardness, three samples were prepared in the same condition, and five data were obtained from each sample. For the tensile test, three samples were prepared in the individual condition.

## 4. Conclusions

This study explored the influence of Mn addition and two-stage homogenization heat treatment on the Al–7Mg–0.15Ti (B535.0) alloy by observing the microstructure of the alloy and testing the mechanical properties. The results of the study are summarized as follows:The Al–7Mg–0.15Ti (B535.0) alloy had good hot plastic and cold plastic workability, and it was successfully made into a forged alloy.After the homogenization heat treatment of the Al–7Mg–0.15Ti alloy with the addition of manganese (0.8Mn), the dendritic grains in the as-cast state were fully eliminated. In addition, Al_4_Mn dispersoids precipitated in the aluminum matrix. Compared with the one-stage homogenization, the two-stage homogenization resulted in a precipitated phase that was finer and more densely distributed.The Al_4_Mn dispersoids inhibited the movement of the alloy grain boundaries and refined the crystal grains. As a result, after the two-stage homogenization and annealing, the manganese-containing Al–7Mg–0.15Ti alloy had the finest grains.The addition of Mn to the alloy resulted in the refinement of the as-cast and recrystallized grains, and reduced the conductivity of the alloy. In addition, the cold-rolled Mn-containing alloy that was subjected to the two-stage homogenization had the highest tensile strength.

## Figures and Tables

**Figure 1 molecules-26-00168-f001:**
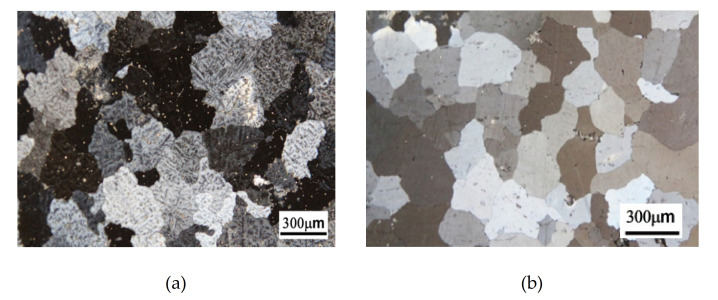
Optical microstructure observations of (**a**) as-cast Alloy A (0Mn); (**b**) 1-Homo Alloy A (0Mn); (**c**) 2-Homo Alloy A (0Mn); (**d**) as-cast Alloy B (0.8Mn); (**e**) 1-Homo Alloy B (0.8Mn); (**f**) 2-Homo Alloy B (0.8Mn). 1-Homo: after the one-stage homogenization; 2-Homo: after the two-stage homogenization.

**Figure 2 molecules-26-00168-f002:**
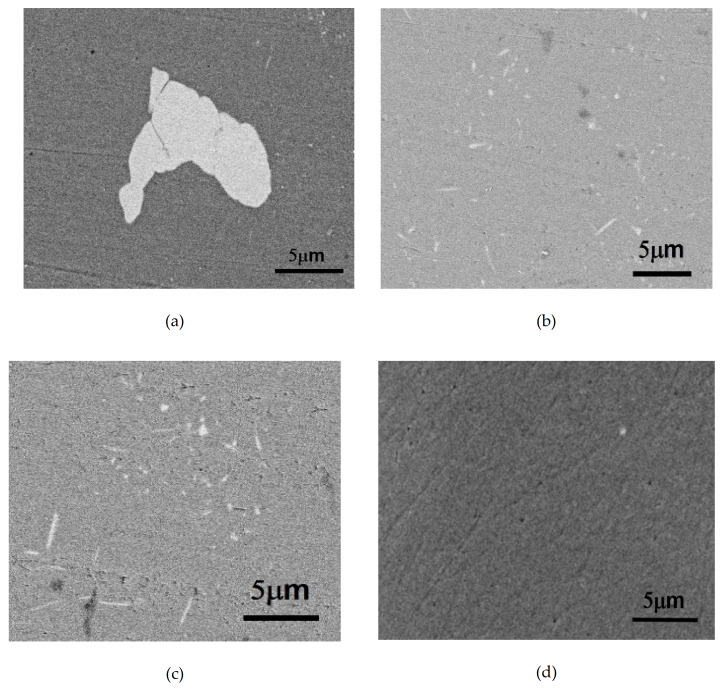
SEM-BEI (Backscattering Electron Image of Scanning Electron Microscope) observation of the Al_6_Mn dispersoids after (**a**) as-cast alloy B (0.8Mn), Al_6_Mn EDS analysis: Al: 70.0 wt.% Mn: 30.4 wt% Mg: 1.1 wt%; (**b**) 1-Homo Alloy B (0.8Mn); (**c**) 2-Homo Alloy B (0.8Mn); (**d**) as-cast Alloy A (0Mn).

**Figure 3 molecules-26-00168-f003:**
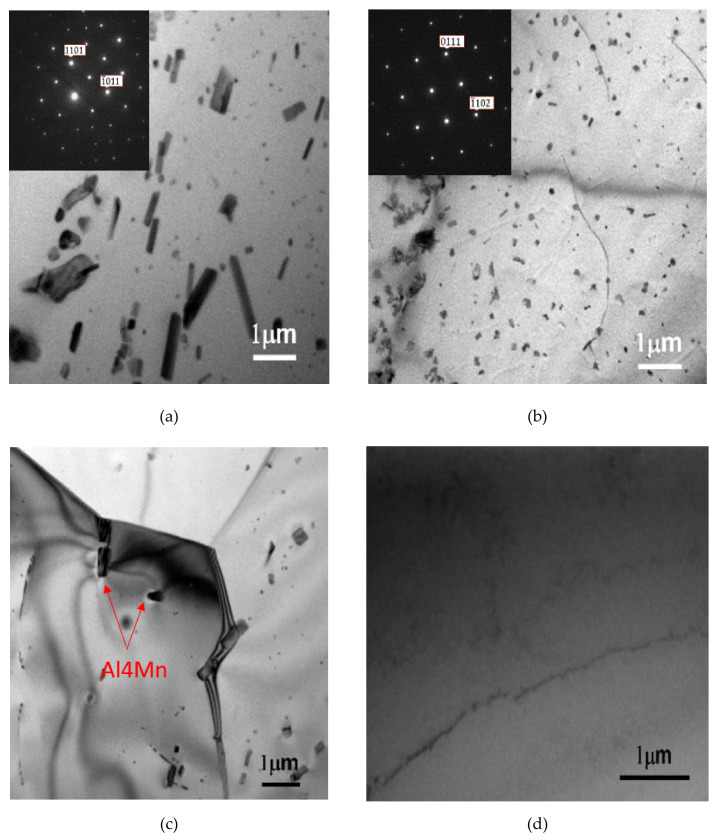
TEM (transmission electron microscopy) observation of the Al_4_Mn and dispersoids in Alloy B (0.8Mn) after (**a**) 1-Homo with [01$\bar{1}$1] zone axis; (**b**) 2-Homo with [5$\bar{1}\bar{4}$3] zone axis; (**c**) Al_4_Mn dispersoids on the grain boundary, (**d**) as-cast Alloy A (0Mn).

**Figure 4 molecules-26-00168-f004:**
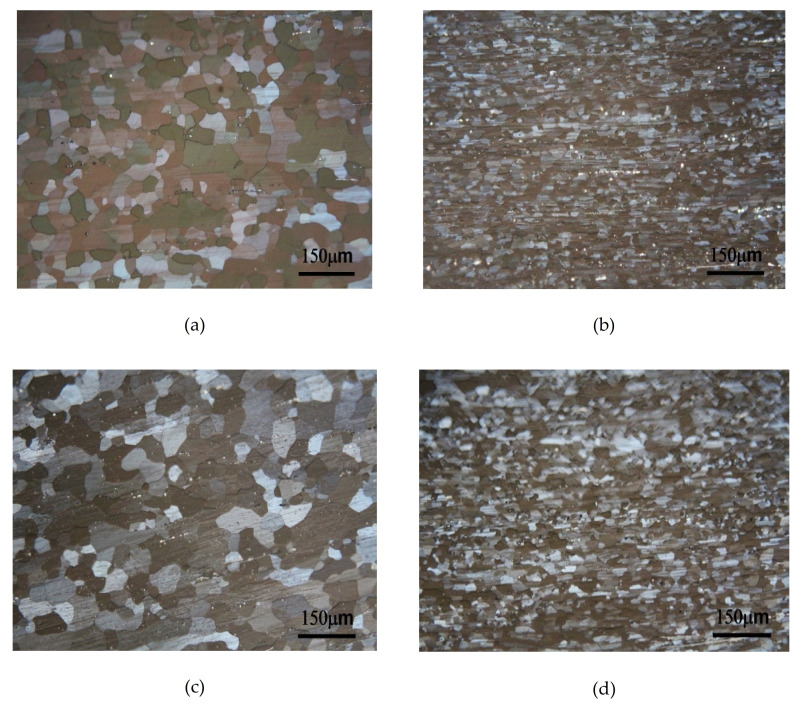
Observation after full annealing (**a**) 1-Homo Alloy A; (**b**) 2-Homo Alloy A; (**c**) 1-Homo Alloy B; (**d**) 2-Homo Alloy B; (**e**) rolling direction.

**Figure 5 molecules-26-00168-f005:**
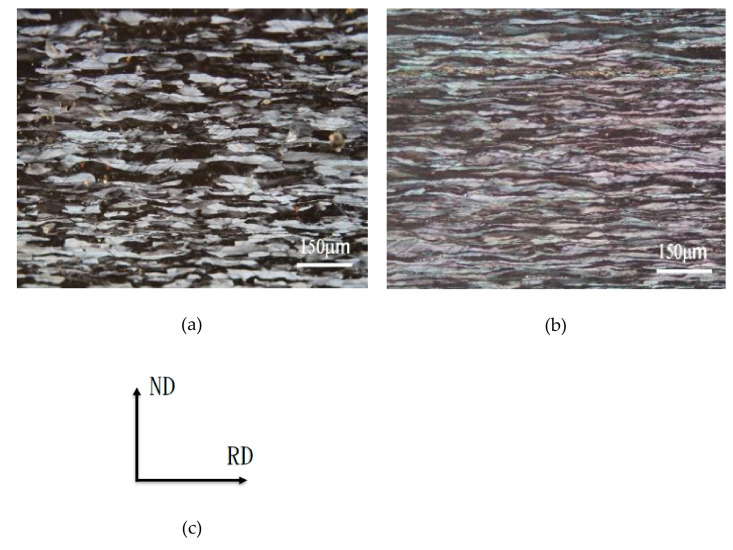
Optical microstructure observation of (**a**) cold rolling Alloy A (0Mn); (**b**) cold rolling Alloy B (0.8Mn); (**c**) rolling direction.

**Figure 6 molecules-26-00168-f006:**
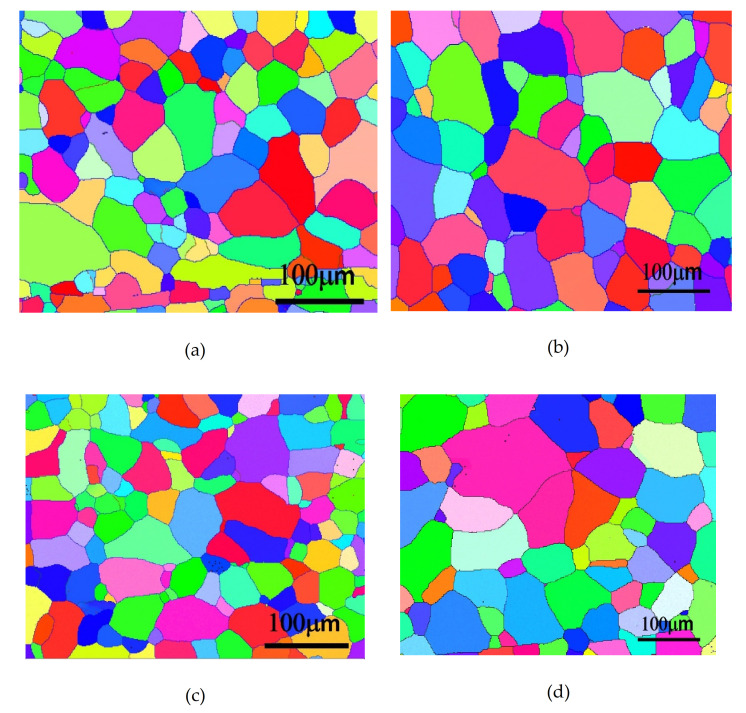
EBSD (Electron Backscatter Diffraction) analysis of Alloy A (0Mn) after (**a**) 1-Homo 400 °C annealing for 0.5 h; (**b**) 1-Homo 400 °C annealing for 1 h; (**c**) 2-Homo 400 °C annealing for 0.5 h; (**d**) 2-Homo 400 °C annealing for 1 H; (**e**) Inverse Pole Figure (IPF) coloring map and rolling direction.

**Figure 7 molecules-26-00168-f007:**
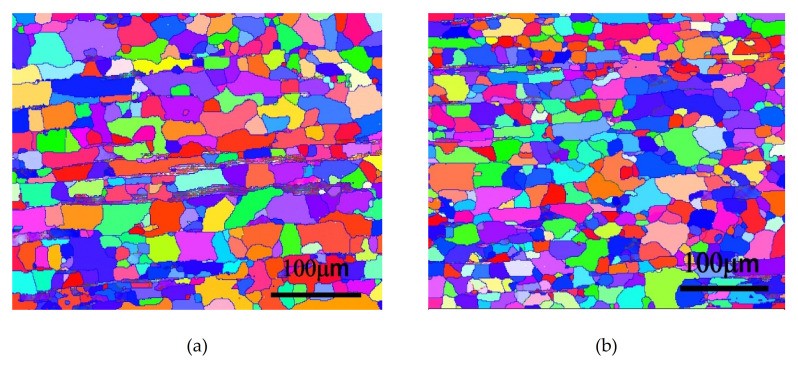
EBSD analysis of Alloy B (0.8Mn) after (**a**) 1-Homo 400 °C annealing for 0.5 h; (**b**) 1-Homo 400 °C annealing for 1 h; (**c**) 2-Homo 400 °C annealing for 0.5 h; (**d**) 2-Homo 400 °C annealing for 1 h; (**e**) IPF coloring map and rolling direction.

**Figure 8 molecules-26-00168-f008:**
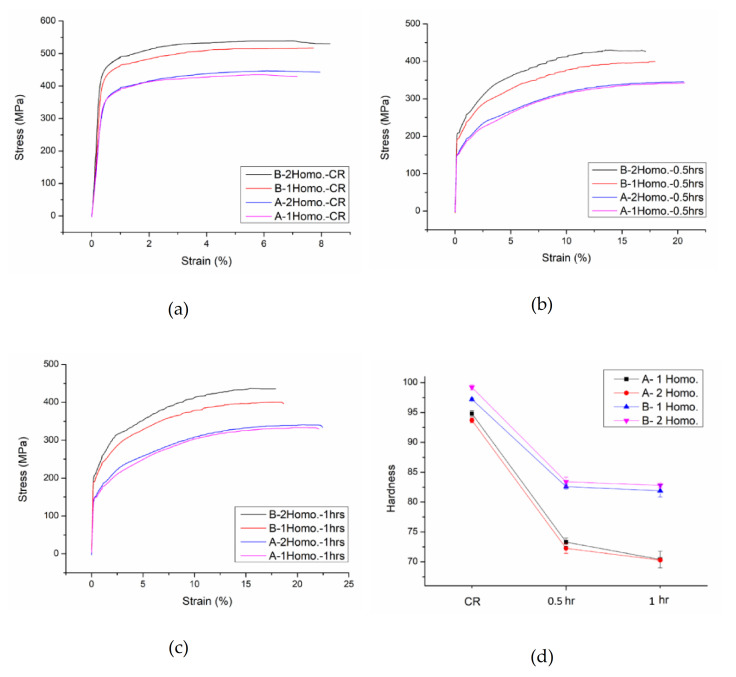
The mechanical properties of Alloy A (0Mn) and Alloy B (0.8Mn): (**a**) tensile curves after cold-rolled; (**b**) tensile curves after 400 °C annealing for 0.5 h; (**c**) tensile curves after 400 °C annealing for 1 h; (**d**) hardness curves.

**Figure 9 molecules-26-00168-f009:**
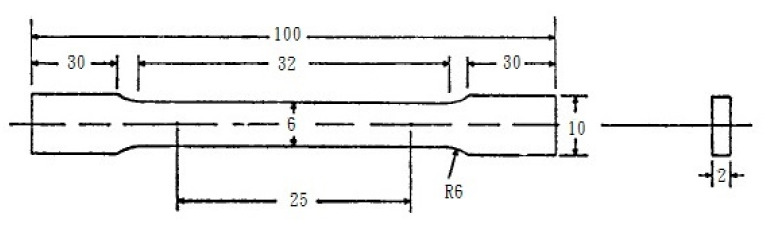
The dimensions of the tensile test piece (unit: mm) [22].

**Table 1 molecules-26-00168-t001:** A summary table of the measurement results of the conductivity (%IACS) of aluminum–magnesium alloys in various processes.

	A (0Mn)			B (0.8Mn)			
State	1-Homo (a)	2-Homo (b)	$\frac{\left(b-a\right)}{a}\times 100\%$	1-Homo (c)	2-Homo (d)	$\frac{\left(d-c\right)}{c}\times 100\%$	$\frac{\left(d-b\right)}{b}\times 100\%$
Cold-rolled	26.49(0.07)	26.36(0.11)	−0.4%	23.33(0.16)	23.32(0.02)	0%	−11.5%
0.5 h annealing	27.41(0.05)	27.38(0.05)	−0.1%	24.24(0.05)	23.94(0.03)	−1.2%	−12.6%
1 h annealing	27.44(0.05)	27.39(0.03)	−0.2%	24.29(0.03)	24.04(0.03)	−0.8%	−12.2%

( ): Standard deviation.

**Table 2 molecules-26-00168-t002:** A summary table of the test results of the mechanical properties of aluminum–magnesium alloys in various processes.

State	Mechanical Properties	A (0Mn)			B (0.8Mn)			
		1-Homo (a)	2-Homo (b)	$\frac{\left(b-a\right)}{a}\times 100\%$	1-Homo (c)	2-Homo (d)	$\frac{\left(d-c\right)}{c}\times 100\%$	$\frac{\left(d-b\right)}{b}\times 100\%$
cold-rolled	HRF	94.8(0.5)	93.7(0.4)	−1.1%	97.2(0.2)	99.2(0.4)	2.1%	5.9%
	YS(MPa)	276.1(1.6)	279.3(2.9)	1.1%	351.5(3.1)	363.3(4.3)	3.3%	30%
	UTS(MPa)	446.0(5.9)	447.5(2.2)	0.3%	518.7(1.7)	539.9(0.4)	4.1%	20.1%
	EL%	7.2(0.9)	7.7(1.7)	6.9%	7.8(0.1)	8.3(0.5)	6.4%	7.8%
0.5 h annealing	HRF	73.3(0.7)	72.3(0.9)	−1.4%	82.6(0.4)	83.4(0.8)	1%	15.3%
	YS(MPa)	146.6(1.2)	143.9(4.3)	−1.8%	191.3(1.8)	197.2(4.7)	3.1%	37%
	UTS(MPa)	345.9(3.8)	348.5(1.5)	0.7%	404.2(2.1)	436.9(1.9)	8.1%	25.3%
	EL%	20.5(2.1)	20.5(1.9)	0%	17.9(1.4)	17.4(0.7)	−2.7%	−15.1%
1 h annealing	HRF	70.4(1.4)	70.3(0.2)	−0.1%	81.9(1.0)	82.8(0.4)	1%	17.8%
	YS(MPa)	137.8(1.3)	137.4(0.8)	−0.2%	190.7(1.1)	195.1(5.7)	2.3%	38%
	UTS(MPa)	339.7(4.9)	340.9(0.9)	0.3%	403.3(1.9)	435.5(1.2)	8%	27.7%
	EL%	22.1(0.7)	22.5(1.1)	1.8%	18.2(0.4)	17.9(0.2)	−1.6%	−20.4%

( ): Standard deviation.

**Table 3 molecules-26-00168-t003:** Alloy Composition (wt.%).

Composition Alloy	Mg	Mn	Ti	Fe	Si	Cu	Cr	Al
Alloy A (0Mn)	7.09	<0.01	0.15	0.10	0.05	<0.01	<0.01	Bal.
Alloy B (0.8Mn)	6.95	0.78	0.14	0.11	0.04	<0.01	<0.01	Bal.

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
