# Peer review of "The Effects of a Trace Amount of Manganese and the Homogenization on the Recrystallization of Al–7Mg–0.15Ti Alloys"

_molecules, 2020, doi:10.3390/molecules26010168_

Round 1
Reviewer 1 Report
In the article, the authors presented the effects of the Mn addition and the homogenization treatment on the microstructures and mechanical properties of the Al-7Mg-0.15Ti alloy.
The experiments are well-conducted and written well. I recommend publishing after minor revisions.
Here are my reviews regarding this work,
- In the introduction, some sentences are too long, difficult to understand.
FOR EXAMPLE,
One of the commonly used fibers related to the enhanced biodegradability of PP composites is wood fibers (WF).
Most sentences may be unclear or hard to follow. Consider rephrasing in many places.
- CHECK THE ENTIRE MANUSCRIPT WITH native English speaker.
Author Response
Response to Reviewer 1’s Comments
Dear Reviewer:
Thank you for your precious comments concerning our manuscript entitled “The Effects of a Trace Amount of Mn and the Homogenization on the Recrystallization of Al-7Mg-0.15Ti Alloys” (ID: molecules-1024359). Those comments are all valuable and very helpful for revising and improving our paper, as well as the important guiding significance to our researches. We have studied the comments carefully and made corrections which we hope will meet with your approval. The modifications and supplements are marked in red in the revised paper. The main corrections in the paper and the responses to the comments are as follows:
Reviewer1
In the article, the authors presented the effects of the Mn addition and the homogenization treatment on the microstructures and mechanical properties of the Al-7Mg-0.15Ti alloy.
The experiments are well-conducted and written well. I recommend publishing after minor revisions.
Here are my reviews regarding this work
POINT 1. In the introduction, some sentences are too long, difficult to understand.
For example,
One of the commonly used fibers related to the enhanced biodegradability of PP composites is wood fibers (WF).
Most sentences may be unclear or hard to follow. Consider rephrasing in many places.
Check the entire manuscript with native English speaker.
Response:
Thank you for your compliments and suggestions. The grammar and terminology of the manuscript has been revised by English experts (MDPI English Editing Services, ID: english-25241). The reviewer is sincerely requested to review it again.
It is considerate of you to provide such detailed guidance. We sincerely hope that our responses will be satisfactory. Please inform us if you have any further comments or suggestions on the manuscript.
Special thanks for your constructive comments.

Reviewer 2 Report
In the paper the Authors investigate the effects of the trace amount addition of the Mn to B535.0 alloy as well as thermal treatment on its microstructure and mechanical properties.
The paper is well-organized but more focus should be put on graphical aspects ( i.e. scales in the figures are given in 3 different manners – white text, black text and black text on the white box; font size changes; data from Table 3 would be more clear to analyze if given as a graph; in line 183 there is reference to Fig. 4d which is not present in the manuscript)
The paper can be accepted if the authors will refer to the following remarks, provide additional information and do the necessary corrections, which would further improve the paper:
- In Introduction, please provide the information on other instability criteria than Considere (https://doi.org/10.1016/j.scriptamat.2013.12.009) and justify your choice
- In Materials and Methods, provide the information on the hardness test depth as well as the side of the test on the specimen concerning the rolling direction, how many samples were prepared? How many tests were carried out (hardness and tensile)?
- In Results and Discussion, before the treatment grain size is said to be over 100 μm for sample 2 mm thick which make only a few grains in the cross-section, how can you prove that the change in the properties is not just a specimen size effect, especially for Elongation? (https://doi.org/10.1016/j.msea.2011.11.021)
- For Fig3d, could you provide the SEM micrograph in at least the same scale as the other images in Fig. 3? Maybe dispersoids are smaller and not clearly visible in such scale.
- Fig3 is used to show existing phases from SEM-BSE, could you confirm these findings with XRD? May it also be used to confirm the grain sizes?
- Please provide typical stress-strain as well as hardness tests curves.
Author Response
Response to Reviewer 2’s Comments
Dear Reviewer:
Thank you for your precious comments concerning our manuscript entitled “The Effects of a Trace Amount of Mn and the Homogenization on the Recrystallization of Al-7Mg-0.15Ti Alloys” (ID: molecules-1024359). Those comments are all valuable and very helpful for revising and improving our paper, as well as the important guiding significance to our researches. We have studied the comments carefully and made corrections which we hope will meet with your approval. The modifications and supplements are marked in red in the revised paper. The main corrections in the paper and the responses to the comments are as follows:
Reviewer2
In the paper the Authors investigate the effects of the trace amount addition of the Mn to B535.0 alloy as well as thermal treatment on its microstructure and mechanical properties.
POINT 1. The paper is well-organized but more focus should be put on graphical aspects ( i.e. scales in the figures are given in 3 different manners – white text, black text and black text on the white box; font size changes; data from Table 3 would be more clear to analyze if given as a graph; in line 183 there is reference to Fig. 4d which is not present in the manuscript)
The paper can be accepted if the authors will refer to the following remarks, provide additional information and do the necessary corrections, which would further improve the paper:
Response:
Thank you for your kind reminder. The scale and font size of each picture has been adjusted. Figure 4(d) has been supplied.
POINT 2. In Introduction, please provide the information on other instability criteria than Considere (https://doi.org/10.1016/j.scriptamat.2013.12.009) and justify your choice
Response:
Thank you for providing the article on the instability criteria. Your suggestion is illuminating, for our team is currently researching the Ultrafine Grain Al alloy. We will add Hart's criteria to Line 75.
POINT 3. In Materials and Methods, provide the information on the hardness test depth as well as the side of the test on the specimen concerning the rolling direction, how many samples were prepared? How many tests were carried out (hardness and tensile)?
Response:
For the test of hardness, three samples were prepared in the same condition, and five data were obtained from each sample. For the tensile test, three samples were prepared in the individual condition, as shown in Line 124 - 126
POINT 4. In Results and Discussion, before the treatment grain size is said to be over 100 μm for sample 2 mm thick which make only a few grains in the cross-section, how can you prove that the change in the properties is not just a specimen size effect, especially for Elongation? (https://doi.org/10.1016/j.msea.2011.11.021)
Response:
Thank you for providing the article on the influence of thickness on elongation, which is very helpful for our subsequent research. We also start to study the characteristics of the Al-Mg alloy in the literature, but find that the results of the current experiment in the paper cannot prove whether the size of the test piece can affect the ductility or not. However, we will add the article that you provided to the reference. We will continue to combine simulation and implementation in the future. Thank you very much for your ideas.
POINT 5. For Fig3d, could you provide the SEM micrograph in at least the same scale as the other images in Fig. 3? Maybe dispersoids are smaller and not clearly visible in such scale.
Response:
We have already provided a new picture for Figure 3 in Line 169-171. Under the smaller magnification, the dispersoids are still invisible.
POINT 6. Fig3 is used to show existing phases from SEM-BSE, could you confirm these findings with XRD? May it also be used to confirm the grain sizes?
Response:
We will subsequently use the EDS analysis (LINE 169) to confirm the dispersoids.
If we simply use SEM, it may not be able to confirm the grain size. Although the grain size can be observed by etching the test piece, the dispersoids will disappear at the same time. Therefore, SEM is mainly used to observe the dispersoids instead of the grain size.
POINT 7. Please provide typical stress-strain as well as hardness tests curves.
Response:
We have provided the test curves in Figure 9. Thank you very much.
It is considerate of you to provide such detailed guidance. We sincerely hope that our responses will be satisfactory. Please inform us if you have any further comments or suggestions on the manuscript.
Special thanks for your constructive comments.

Round 2
Reviewer 2 Report
The paper can be accepted in the current form.